# A Review of Flow Characterization of Metallic Materials in the Cold Forming Temperature Range and Its Major Issues

**DOI:** 10.3390/ma15082751

**Published:** 2022-04-08

**Authors:** Man-Soo Joun, Mohd Kaswandee Razali, Chang-Woon Jee, Jong-Bok Byun, Min-Cheol Kim, Kwang-Min Kim

**Affiliations:** 1Engineering Research Institute (ERI), School of Mechanical and Aerospace Engineering, Gyeongsang National University, Jinju 52828, Korea; mohdkaswandeerazali@gnu.ac.kr (M.K.R.); jbbyun@gnu.ac.kr (J.-B.B.); 2Technical Research Laboratories, POSCO, Gwangyang 57807, Korea; wirehog@naver.com; 3Metal Forming Research Corporation (MFRC), Research Center, Jinju 52818, Korea; mckim@afdex.com; 4Sungjin Foma Co., Ltd., Daegu 42709, Korea; kimmen99@naver.com

**Keywords:** flow characterization, room temperature, review

## Abstract

We focus on the importance of accurately describing the flow behaviors of metallic materials to be cold formed; we refer to several valuable examples. We review the typical experimental methods by which flow curves are obtained, in addition to several combined experimental-numerical methods. The characteristics of four fundamental flow models including the Ludwik, Voce, Hollomon, and Swift models are explored in detail. We classify all flow models in the literature into three groups, including the Ludwik and Voce families, and blends thereof. We review the experimental and numerical methods used to optimize the flow curves. Representative flow models are compared via tensile testing, with a focus on the necking point and pre- or post-necking strain hardening. Several closed-form function models employed for the non-isothermal analyses of cold metal forming are also examined. The traditional bilinear *C-m* model and derivatives thereof are used to describe the complicated flow behaviors of metallic materials at cold forming temperatures, particularly in terms of their applications to metal forming simulations and process optimization.

## 1. Introduction

Material characterization [1,2,3] has become increasingly important since the accuracy of finite element (FE) predictions is dependent on flow behaviors [4] and tribological factors [5]. Important topics in metal forming engineering involve the solution accuracy of FE predictions to innovate optimal process design technology [6], encountering strong demand from industry. During bulk metal forming, flow stress data at larger strains, for example, over 3.0 in automatic multi-stage cold forging [7,8,9], are required, in addition to data on how cumulative damage affects the flow stress and fracture risk [10,11]. Accurate flow behaviors are thus of great importance.

Many researchers [12,13,14,15,16,17,18,19,20,21,22,23,24,25,26,27,28,29,30,31,32] studied the acquisition methods of flow behaviors at a high strain at room temperature, including using direct experimental methods, the empirical or analytical correction factor method [12,13], the force-area measurement method [14] and the inverse method [15,16,17,18,19]. There are various test methods for flow characterization such as the cylindrical tensile test, sheet specimen tensile test, tension/compression test [20], sheet bulging test [20,21], layer compression test [22], multi-layer plane strain compression test [23], in-plane torsion test [24,25], torsion test [26], shear test [27], forward extrusion test [28], cruciform–biaxial test [29], rolling test [30], hydraulic bulge test [31], etc. However, many researchers are still relying on the traditional tensile test-based extrapolation method, especially in bulk metal forming engineering, which may be quite erroneous at high strains in most cases [33,34,35]. This is owing to complexity, cost, and lower reliability of the developed methods. It is noted that most test methods were aimed to be applied to sheet materials, and the experimental flow constants including the strain hardening exponent vary greatly between the test methods.

The tensile and cylinder compression tests are the most common. When the simple compression test is employed to obtain the flow stress, we can obtain reliable flow stress only at the relatively low strain due to the strain non-homogeneity inherently occurring in the solid cylinder compression test. The authors believe, based on experience, that the reliable strain limit of this test is near or less than 0.5 because of the frictional effect and high sensitivity of the plastic deformation region to the stress state [36], even though Hering et al. suggested a larger strain limit of 0.7 for this test.

The tensile test is the most attractive because it does not only involve a considerably deformed region but it also accompanies the fracture phenomena. Necking phenomena in the tensile test made it difficult to conduct research works on material characterization at high strain using the tensile test even though a few researchers tackled this matter to obtain some meaningful results using the measured [12] or predicted [37] necking phenomenon itself. Bridgman [12] measured the curvature at the necking region, which could provide some information about the macroscopic features that could be employed for obtaining post-necking strain hardening characteristics [38,39,40]. However, it can be meaningful only when the flow curves are normal, i.e., monotonically increasing, such as those of cold forgeable carbon steels [4,41,42]. The conventional approaches of obtaining flow stress behaviors at high strain and room temperature cannot be applied to some ultra-high strength steels [43,44], stainless steels [8,45,46], special heat-treated steels [47], aluminum alloys [48,49], new materials [50,51], etc.

On the contrary, the research works characterizing the unique flow behaviors at higher strain were begun due to the need for weight reduction in automobiles, full material information for crack or crash simulation, development of measurement system technologies including DIC [52,53,54,55,56,57,58,59], etc. Recent academic publications in this field [55,56,57,58,59] show that the strength of the related research activities is increasing to reveal thermo-mechanical flow behaviors of new materials with the assistance of test equipment and characterization technology.

With advances in measurement technology, some practical methods for obtaining the flow stress at high strain have been developed, which coupled the experiments with FE predictions. From the mid-1990s, the finite element method (FEM) was studied to characterize the flow behaviors. Inverse technologies [60,61] were revived and advanced gradually. In the late 2000s, Joun et al. [16] presented a practical method for obtaining the flow stress at high strain using the cylindrical tensile test, which can give us the flow stress at the higher strain in the necking region. They could obtain the flow stresses at the strains of 0.9 for ESW105 [4], 1.56 for SCM 435 [4], 1.4 for SWCH10A [62], 1.6 for SUS304 [8] and the like. Our experience suggests that the higher the strain hardening capability, the higher the strain at which the flow stress can be obtained. It is a similar approach to the sheet specimen tensile test, even though it exhibited quite a great numerical effect relative to Joun et al.’s approach to the cylindrical tensile test of solid material. Nonetheless, flow stress behaviors at room temperature or at a cold forming temperature are still being unveiled in some viewpoints. These behaviors are more complicated than those at elevated temperatures [1,3,63,64] not only because of their differences and high correlation among the state variables including strain, strain rate and temperature, but also because of the difficulties in achieving experimental data, especially at higher strains.

These environments did not allow the researchers to find a general rule, compared to the cases in the elevated temperature. Even materials with the same chemical composition may behave in quite different ways macroscopically, depending on the prework or heat treatment. For example, the pre-heat-treated steel, ESW steel coil [4,65], is quite different from its original steel with the same chemical composition. Their high correlation with the prework or heat treatment prohibited the researchers from formulating the flow behaviors at room temperature as a separable variable function of strain, strain rate, and temperature, which has been traditionally applied to those at the elevated temperature. These diversities and high correlation arise from high dependence of flow behaviors on the history of plastic deformation and metallurgical properties.

Many researchers have been trying to invent closed-form function flow models. Traditionally, typical true stress-strain curves of the materials to be cold bulk metal formed, i.e., their flow stress curves at room temperature were described by Ludwik [66], Voce [67], Hollomon [68], and Swift [69] in their flow models. Many good material models [70,71,72,73,74,75,76,77,78,79,80,81,82,83,84,85,86,87,88,89,90,91,92,93,94,95,96,97,98] were developed for satisfying their specific situations and applications. Notably, most material flow models at room temperatures were developed based on the fundamental flow models represented by the Ludwik (including Hollomon and Swift models) and Voce models. These are true of the flow models at the elevated temperature. For example, models by Fields-Bachofen [70], Zhang [72], Chen et al. [77], Lin and Chen [78], Johnson-Cook [79], Khan-Huang-Liang [80,81], Ludwigson [82], Zerilli and Armstrong [83], Voyiadjis-Almasri [84], Bodner-Partom [85], Nadai and Manjoine [86], and Joun et al. [16] can be considered to be extended Ludwik models. On the other hand, models by Hartley-Srinivasan [87], Kim-Tuan [88], Rusinek-Klepaczko [89] and Ghosh [30], and models by Voce-Kocks [91], Double-Voce [18], Hockett-Sherby [92], modified Hockett-Sherby [93], El-Magd et al. [94], Voce generalized [95], Bergstorm [96] and Voyiadjis-Abed [97] are classified as extended Swift (represented by Ludwik model) and Voce models, respectively. Therefore, a detailed understanding of fundamental flow models with their characteristics is important. Please note that the original Arrhenius [98] flow model does not involve any strain terms and we thus neglected it in this reviewing study on flow characterization at high strains.

Among the aforementioned models, most flow models [70,79,80,81,82,86,87,88,89,90,91,92,93,94,95,96,97,98] cover flow behaviors with respect to the strain rate and temperature. In most FE analyses of cold metal forming processes, temperature and strain rate effects have traditionally been neglected. However, the flow stress dependence on temperature is important, especially in automatic multi-stage cold forging because of high speed and serial production. The strain rate effects cannot be also neglected especially at lower temperature.

Dynamic strain aging [99,100,101,102,103] exhibits one of special flow behaviors at the cold forming temperature of metals. Voyiadjis et al. [99] created a constitutive model of DSA based on the mechanism associated with the evolution of dislocation density. Song et al. [100] assumed that the overall response of a material was the product of thermal, thermally activated, and DSA stress components when modeling flow stress. Song and Voyiadjis [101] developed a probability density function-based constitutive model of DSA for HCP metals. Joun et al. [102] showed that the dynamic strain aging phenomena of the S25C steel can be effectively described by the traditional piecewise bilinear function model, i.e., the *C-m* model. They also presented a closed-form function model [103] for the S25C, which can be extrapolated to cover the state variables in the wide range of cold forming.

In this study, we first conduct a case study on typical flow behaviors of commercial materials at the temperature range of cold forming. After emphasizing the importance of a detailed understanding of tensile test together with a specific flow model for some insight and knowledge on macroscopic and metallurgical phenomena including even fracture [104], we review the flow models, classify them in Ludwik and Voce flow model families, and evaluate them in terms of tensile test with an emphasis on predicted necking points using FEM. The coverage of this paper includes only the material modeling for conventional cold forming processes. The fundamental flow models [66,67,68,69] are evaluated using the material of SWCH10A and its flow stress information [16], which was based on the flow stress identification algorithm from a tensile test of cylindrical specimen. The material models that can be potentially employed for non-isothermal analyses of cold metal forming processes are also introduced, with an emphasis on the piecewise bilinear *C-m* model. Please note that all the FE solutions in this study were obtained using the commercialized rigid- or elastothermoviscoplastic FE software AFDEX (Altair APA) [105,106]

## 2. Strain, Strain Rate and Temperature Effects

### 2.1. Strain Effect

Eom et al. [4] compared the true and engineering stress-strain curves of SCM435 and ESW105, as shown in Figure 1, which were obtained using Joun et al.’s method [16]. The former is a typical strain hardening material and the latter is an especially pre-heat-treated steel which has high strength and negligible strain hardening capability, i.e., almost perfectly plastic material. Using the two flow stresses, Eom et al. simulated the tensile tests. The comparison of engineering stress-strain curves between the predictions with experiments on the two materials in Figure 1 revealed that they are in excellent agreement with each other, excluding at low strain. Please note that the error at low strain was caused by the usage of the Hollomon model in the pre-necking region, which can be enhanced using Swift [41] and the like.

Eom et al. [4] also compared the two materials experimentally and numerically using a forward extrusion process in Figure 2, revealing differences not only in the contact interface but also in the shape of the extruded end. It was shown by Joun et al. [107] that these deformation behaviors can result in a distinct difference in terms of the central bursting defect, which is an inherent and decisive defect in forward extrusion.

Figure 3 shows the engineering and true stress-strain curves of SUS304 to be cold forged, presented by Byun et al. [8]. The solid line was obtained using Joun et al.’s approach [16] and the dotted line was extrapolated to meet the solid line. Similar flow behaviors can also be found in a work by Kweon et al. [45]. This is called double strain hardening, implying that the flow behaviors cannot be described by a simple closed-form function. Almost the same flow behaviors of double strain hardening can be found in the study by El-Magd et al. [94] on a steel CK45N at room temperature.

Prework of materials has a great influence on the flow behaviors of the material especially at room temperature. For an example, when a drawn rod of material SCM435 was upset, the material exhibited quite different flow behaviors from the original materials, as is shown in Figure 4. The material lost its feature of considerable strain hardening in the early strain, depending on the reduction of area. Hering et al. [28] also showed almost the same Bauschinger effect from tensile test of the upset materials, as shown in Figure 5.

### 2.2. Strain Rate and Temperature Effects

An example emphasizing the importance of temperature effect can be found in Figure 6. Notably, a sharp drop in the flow stress of SUS304 occurs with an increase in temperature in its cold forming range whereas the drop is delayed in SUS440C and SUS430 around the temperature at which DSA occurs, at a strain of 1.0. Notably the flow stress of SUS304 drops from 1300 MPa at 0 °C to 730 MPa at 400 °C and its temperature during cold forging can increase to over 400 °C [8], which causes a great mechanical difference. Please note that the absolute value does not have a quantitative meaning because it may change according to the prework depending on the application, but that it is qualitatively meaningful.

An example presented by Byun et al. [8], shown in Figure 7, emphasizes the great difference in die effective stress between isothermal and non-isothermal analyses of a cold forging process. Figure 7a can give us a possibility of cold forging of the SUS304 material if the die and process are optimized. On the other hand, it may be concluded from Figure 7b that the forging of this material is impossible because of high die stress.

Figure 8 shows the changes in flow stress of S25C [102] with the strain rate at temperatures between 100 °C and 400 °C and at the strain of 0.4, indicating that the strain rate has a non-negligible effect on flow stress in cold forming. However, this decreases drastically as the temperature rises. In particular, the strain rate effect of Figure 8 is monotonic, and can thus be formulated by a simple function. Figure 8 shows the different tendencies of strain rate effect on flow stress in the temperature range of cold forming from that at the elevated temperature. In other words, the strain rate effect over 10/s reduces quickly and the strain rate effect becomes weaker as the temperature increases. Please note that the effect of the strain rate is maintained even at the higher strain rate and at the low temperature.

Figure 9 shows an extreme case of the effect of strain rate on plastic deformation of cold forgeable material SWCH10A at room temperature. We tested a shearing process of the 12 mm diameter coiled rod with different forming velocities, revealing that normal (60 rpm) and higher velocity produced a billet with good quality while the lower velocity left the evidence of ductile fracture on the shear surface, as can be seen in Figure 9a,b, respectively. This example amplifies the effect of strain rate of forgeable materials on plastic deformation and fracture phenomenon at room temperature.

Figure 10 shows the variation in flow stress of S25C with temperatures for various sample strain rates and strains, presented by Joun et al. [102]. The solid lines are experimental and the dashed lines were extrapolated. Flow characteristics typical of DSA can be seen and flow stress clearly varied with temperature. Specifically, the flow stress decreased to its lowest point between 200 °C and 300 °C, before increasing to its local maximum. The temperatures of local extremum points depended not on the strain rate, but rather on the temperature. It is noted that these kinds of complicated flow behaviors can be described by the traditional piecewise bilinear *C-m* model. However, Byun et al. [8] and Lee et al. [103] stated that a sort of approximation should be needed for the state variables outside of the ranges defined by the sample state variables of strain and temperature. In this context, the *C-m* model is not advantageous.

Figure 11 shows some selected strain-flow stress curves of Al6082-T6, presented by Yoo et al. [108], showing that flow stress varied considerably with temperature even in the range of 25–300 °C. It can be also seen from Figure 11 that the strain rate increases from 0.1/s to 20/s brought the flow stress increase by 6% (25 °C) to 30% (300 °C) at the strain of 0.1 and by −1% (25 °C) to 30% (300 °C) at the strain of 0.5. The revere effect of strain rate on flow stress should be noted, i.e., strain rate softening occurred when the strain exceeds critical values of strain that increase with temperature.

Please note that peak flow stresses varied with temperature, showing that the flow stress dropped by 50–60% as the temperature increased from room temperature to 300 °C and that the drop rate of flow stress with temperature increased drastically when the temperature reached around 200 °C, called critical temperature of flow stress drop. The strain softening as well as strain rate instability was distinctly observed in the case of Al6082-T6.

It is interesting to note that the effect of strain rate on flow stresses in around the strain range over 0.35 to 0.40 at relatively low sample temperatures of 25 °C and 100 °C became reversed, as stated above. It is also noted that the flow stresses at higher sample strain rates of 10/s and 20/s started to drop drastically, compared to the other cases. The higher strain rate causes the higher drop rate of flow stress except the case of 300 °C. It is not easy to obtain the acceptable flow stress at higher strain, for example, 0.5 from the compression test of solid cylindrical specimens because of their barreling phenomena or non-uniformity of plastic deformation [102].

The instability of plastic deformation was observed in well-lubricated upsetting of a simple aluminum ring at room temperature of which the characteristics can be represented by low strain hardening capability and low melting temperature, resulting in a non-symmetric shape with respect to central cross-section, as shown in Figure 12a. We could predict the unbalanced shape with different frictional condition (coefficient of friction of upper die = 0.10; that of lower die = 0.12) which did not allow normal strain hardening materials, such as SCM435 or SCM415, to experience such a remarkable deformation unbalance.

In Figure 12b, a comparison of the cylinder compression tests (Initial height = 15 mm, initial diameter = 10 mm) also shows that Al6082-T6 exhibited greater instability of plastic deformation, i.e., bad concentricity and roundness of the material in solid cylinder upsetting at room temperature, compared to that of SCM415H. These phenomena can incur inaccuracy of numerical solutions when the causes are not dealt with properly in the analysis model.

## 3. Acquisition of Flow Curve at High Strain

### 3.1. Experimental Methods

There have been many experimental methods used in acquiring the flow behaviors over broad ranges of strain, strain rates, and temperatures [1,2,3,12,13,14,15,19,20,21,22,23,24,25,26,27,28,29,30,31,32,33,34]. The tensile test of cylindrical specimens has many strong points in terms of flow characterization of solid materials especially for cold bulk metal forming engineering. It is simple and well-established as well as very reliable and reproducible. It not only provides major material properties including the Young’s modulus, yield strength, tensile strength, elongation, toughness, etc., but it also provides some implicit information about strain hardening or strain softening behaviors at quite high strains (greater than 0.5) during plastic deformation of the ductile material. Please note that cylindrical specimens in cylindrical tensile tests maintain geometric symmetry even in the region of localized necking, implying that they can experience much greater plastic deformation, which can be used to identify flow stress at high strain and to reveal fracture mechanism of the materials to be metal formed.

Until the late 2000s, when practical the experimental-numerical combined method (ENM) was started for practical application to the standard cylindrical tensile test [16], relatively few researchers [12,109,110] employed it to reveal the room temperature flow behaviors of commercial metals at high strain because of difficulty in dealing with necking. Thus, the material identification from tensile test of cylindrical specimens needs extreme accuracy and numerical robustness [37] because of its high dependence on the necking point, complicated non-linearity, and some numerical schemes [18], which have been the factors discouraging to researchers. Because of this numerical problem, a few researchers may still think that the cylindrical specimen tensile test cannot give us the post-necking flow behaviors, as Hering et al. [28] indicated in Figure 13, even though it is, of course, true when the ENM is excluded. It should be, however, noted that one strong point of material characterization using the tensile test is the possibility of revealing the mechanism of damage accumulation and fracture in the region of localized necking [41,104].

Simple and practical methods for flow characterization at the cold forming temperature also include the cylinder compression test. Even though Hering et al. overestimated the limit strain of a cylinder compression test in Figure 13, its reliable strain limit is less than 0.5 owing to the frictional effect during the test [102]. As the necking is sensitive in the tensile test, the friction in cylinder compression test is very influential. It also directly affected the deformation of tools and experimental instruments. Thus, the cylinder compression itself is not so much powerful for the characterization of flow behaviors at high strains at the cold metal forming temperature.

As summarized in Table 1, many researchers studied various methods of obtaining flow information at high strains, excluding the mere cylinder compression test and standard cylindrical tensile test.

Bridgman [12] made experimental efforts and tried to measure the necking profile of simple tensile test specimen. It was practically applied to analytically correct flow behaviors of materials [38,39,40]. However, the related experiments should be carefully conducted and their results are more or less inaccurate [13]. Mirone [13] thus presented a material-independent solution of the necking problem which was used to calculate the equivalent stress-strain curve at the post-necking strain. Mirone improved Bridgman’s problem of directly measuring the curvature but the method was inevitably exposed to some non-negligible error because no error minimization algorithm was used.

The traditional experimental approaches were often replaced by the digital image correlation (DIC) techniques [52,53,54,55,56,57,58,59] that freed the researchers from the difficulty in measuring the geometric features such as the curvature of the necking profile. Even though the DIC techniques have many strong points, including acceptable accuracy, flexibility, and applicability, the analytical or numerical approaches cannot be underestimated if their accuracy is superior to the experimental methods [16,37].

Hering et al. [28] presented a method for obtaining flow stress at the high strain of 1.7 using the forward extrusion process and tensile test, as conceptually shown in Figure 13. However, the frictional effects are hard to be separated from the results. Zhuang et al. [30] presented a coupled method for multi-pass rolling and tensile test to obtain the flow stress at the moderate strain of sheet material. Bouvier et al. [27] used simple shear tests of rolled metal sheets and obtained flow stresses at the strain of around 0.5 with an emphasis on strain-induced plastic anisotropy.

Smith et al. [21] presented a hydraulic bulge test approach to characterize the flow curve for the post-necking strain hardening behaviors of sheet metals and they could obtain the flow stress at the strain of around 0.55 using the volume measurement method. In their approach, the flow curve at low strain was obtained from tensile test and considerable computed data scatterings were observed. Merklein and Kuppert [22] used the layer compression test of a sheet metal to characterize its flow behavior at both pre- and post-necking strains of up to around 0.45. Kopp and Putten [23] used a miniaturized multi-layer plane strain compression test to characterize the flow curve considering the size effect in the rolling of a very small and thin strip. They could obtain the flow curve at the strain of 0.8. Marciniak and Kolodziejski [24] and Tekkaya et al. [25] employed the plane torsion test to determine the flow curve of sheet metal. Tekkaya et al. presented the flow curves at the strain over 0.8. Kuwabara et al. [29] used a cruciform-biaxial test of sheet material to obtain the flow stress at the high strain of around 1.1. Dziallach et al. [33] extrapolated the flow curves at the pre-necking strain obtained by tensile test of sheet material using various flow models. The extrapolated flow curve was compared with an experimental flow curve obtained using the bulge test to characterize the flow curve of the sheet material of DX54 up to the limit strain of 1.0. Matin et al. [14] measured the cross-section at the necking point and tensile load during tensile test of an aluminum alloy sheet specimen and calculated the flow stress at the strain of around 1.5 with quite poor accuracy. Boger et al. [20] conducted a tension/compression test of sheet specimen to study the Bauschinger effect and obtained flow information at the stain of over 0.2. Lach and Pöhlandt [26] studied an optimized torsion test to characterize the flow behaviors with respect to strain and strain rate. Hwang et al. [31] proposed a hydraulic bulge test assisted by an analytical model using tube thickness at the pole, bulge height and internal pressure and could obtain the flow stress of SUS409 up to the strain of 0.22.

In addition to the above test methods, there are several methods [32] which focus on more specific situations, including biaxial compression tests, biaxial tension tests on metal sheets and tubes using closed-loop electrohydraulic testing machines, the abrupt strain path change method for detecting a yield vertex and subsequent yield loci without unloading, in-plane stress reversal tests on metal sheets, and multistage tension tests.

We summarized the test or flow characterization methods in Table 1 together with the several inverse methods [15,16,18,19]. It can be seen that the tensile test of standard cylindrical specimens is relatively rare to other test methods even though it is simple and economical. The comparison in Table 1 reveals that the inverse methods based on tensile test and FEM are the engineering solutions of flow characterization of metallic materials at high strains.

### 3.2. The Experimental-Numerical Combined Method

Zhang and Li [107] suggested an elastoplastic FE analysis coupled way of predicting the tensile test of a cylindrical specimen, which has a function of modifying flow stress information to trace the given tensile load–elongation curve. Even though their approach is not practical and exposed to a non-negligible amount of error, they belong to the pioneering researchers in this field. Cabezas and Celentano [108] simulated the tensile tests of cylindrical and sheet specimens using the flow stress obtained from the Bridgman correction method, showing that using FE analysis of the tensile test was possible for material characterization even though they did not use the FE analysis technique directly to identify the flow stress. Kajberg and Lindkvist [15] employed an inverse modeling technique, based on DSP-technique and FEM, to characterize the flow curves of hot-rolled thin steel sheets and obtained flow stress up to 0.8 in the case of a ductile steel. Joun et al. [16] presented a practical and general ENM approach based on cylindrical tensile test and FEM, which has a lot of successful applications.

In the 2010s, many researchers studied the post-necking strain hardening phenomena to reveal the plastic deformation behaviors of the material and opened a renaissance of the ENM and application of post-necking hardening to reveal the plastic deformation and fracture after necking. One of the reasons lies in the fact that both the numerical methods and experimental technologies including DIC have been much advanced. It also lies in the demand from the CAE engineers or researchers who are interested in higher accuracy of FE predictions in revealing the fracture phenomena in tensile test of cylindrical specimens or in applying new high strength materials. Literature survey on post-necking strain hardening in tensile test of standard cylindrical specimens shows that this trend has still been growing.

In the early 2010s, Kamaya and Kawakubo [19] presented an ENM approach to characterize the flow behaviors using an axi-symmetric hourglass-type tensile specimen, by coupling DIC technologies and FEM. They could obtain the flow curve at the strain of 0.85 in the case of a carbon steel SM490A with no pre-strain. Ganharul et al. [111] and Donato et al. [112] defined the problem clearly and suggested an engineering measure to give further support to the determination of true material properties considering severe plastic deformation after necking. It is noteworthy that they emphasized properly the importance of ENM especially for experimental techniques including DIC. Kim et al. [17] employed an inverse method to the virtual fields method to characterize the post-necking strain hardening behavior of sheet material using a standard tensile test and DIC technique. Eom et al. [104] coupled various damage models with the identified flow behaviors to evaluate the damage models in terms of fracture prediction. In the mid 2010s, Zhu et al. [113,114] applied the DIC technique to measure true stress-strain curves from tensile test of cylindrical specimen. Majzoobi et al. [115] presented a semi-analytical approach to correct the stress-strain curve for the high strain.

Zhao et al. [18] presented an inverse method for obtaining the flow curves at high strains from the sheet specimens using a simple tensile test combined with its FE analysis. They could obtain the flow curve at the strains around 0.85 of Q195 and 0.3 of Al6061, even though their approach was exposed to some numerical uncertainty including the mesh size effect due to its high sensitivity to mechanics at the necking point. Wang et al. [116] established an ENM minimizing the error between the experimental and predicted load-displacement curves to describe simple tensile test after necking. Jeník et al. [117] suggested two ways, i.e., an ENM and a neural network approach to identify the materials from the tensile test of cylindrical specimens. Kweon et al. [118] studied an ENM based on the cylindrical tensile test with well-known material models and their variants to identify the material in terms of flow stress after necking, revealing that the modified Hollomon law gives better flow stresses of SA-508 Grade 3 Class 1 low alloy steel. Paul et al. [119] suggested a simplified procedure for correcting the post-necking strain hardening behaviors based on DIC experiments with an emphasis on local strain measurement at the necked region. Mirone et al. [120] presented a two-step approach to calculate the flow stress information, i.e., which translates first the engineering curves into their corresponding temporary curves, called true curves, using a material-independent function and then the true curves into the flow curves using the final correction function. They achieved some generality compared to the previous research work [9] but the error in terms of tensile load between new solutions and experiments can still be found clearly in their comparison. Chen et al. [121] studied an ENM based on a tensile test of cylindrical specimen using a modified Voce hardening model together with an optimization technique. They could obtain the flow curve of a mild steel with high accuracy. However, their approach is hardening model-dependent and numerical scheme-dependent and thus has some limitations and computational inefficiency. For example, it cannot deal with the localized necking phenomena and/or it may fail to obtain an acceptable solution within a limited computational time because of its high dependence on the necking phenomena. As indicated by Tu et al. [2], the calculated post-necking strain hardening is much dependent on the material models and that the post-necking strain hardening is thus exposed to some numerical convergence and cost-ineffectiveness.

However, Joun’s approach avoids such problems, as indicated by Tu et al. It characterizes the room temperature flow stresses at larger strain from the post-necking strain hardening phenomena, based on tensile testing of a simple cylindrical specimen using a FEM [37]. They did not use any optimization techniques and the scheme can thus be perfectly separated from the FE program [47]. It employs a modified Hollomon model, which can describe any kind of flow stress functions even for almost a perfectly plastic material [4] because the strength coefficient is formulated by piecewise linear functions defined by the set of sampled strains and their related strength coefficient values to be calculated. Eom et al. [4] showed that the model could accurately calculate flow stress even at the strain of 1.5 in the case of SCM435, yielding an engineering stress-strain curve with negligible errors. In addition, the method can be used to reveal the material deformation behaviors even between fracture start point and fracture point [113], which is important to understand the fracture phenomena occurring in tensile test. Joun et al.’s approach also has a unique and important strong point, in that it can also be used to predict the flow behaviors during softening just before fracture [35]. However, the results are composed of a discrete flow information at the sample points of strain. Our experiences show that maximum strain at the last sample point ranges from 0.9 for ESW105 [4] to 1.6 for SUS304 [8] and a sort of extrapolation is thus inevitable especially for automatic multi-stage cold forging where effective strain may exceed 3.0 or 4.0 at locally deformation-concentrated region [8].

Very recently, Li et al. [53] employed a special multi-camera DIC system to directly determine the true stress-strain curve over a high strain range from the tensile test of sheet specimen, revealing that there exists a distinct softening start point.

## 4. Strain Hardening Models

### 4.1. Fundamental Flow Models in Terms of Tensile Test

In Section 2, we saw various features of flow behaviors of metallic materials in the temperature range of cold forming. Many researchers tried to find strain hardening models to describe the flow behaviors with limited number of material constants. The number of material constants has been one of the important measures in flow models because fewer material constants is more practical and gives some easy insight into the materials’ flow behaviors. In this context, the representative strain hardening models include Ludwik (1909) [66], Voce (1948) [67], Hollomon (1945) [68] and Swift (1952) [69] flow models, which are defined by two or three material constants. They are formulated as a function of strain, as follows:(1)$$\sigma =Y_{0}+L_{1}\varepsilon^{n_{L}}$$
(2)$$\sigma =Y_{0}+V_{1}\left(1-e^{-V_{2}\varepsilon }\right)$$
(3)$$\sigma =L_{1}\varepsilon^{n_{L}}\;\mathrm{or}\;\sigma =K\varepsilon^{n}$$
(4)$$\sigma =Y_{0}{\left(1+S_{1}\varepsilon \right)}^{n_{S}}$$
where $Y_{0}$ is the yield strength of the material, i.e., strain-free flow stress and $L_{1}$ and $S_{1}$ are called Ludwik and Swift model constants, respectively. $K(=L_{1})$ and $n(n_{L})$ are known as the strength coefficient and strain-hardening exponent, respectively. $n_{L}$ and $n_{S}\;$are called Ludwik and Swift strain hardening exponents (or strain hardening exponents, sometimes), respectively.

The most important strain that should be considered when measuring flow stress via tensile testing of a cylindrical specimen is the strain at the necking point where the Considère condition [122] should be satisfied, i.e., the flow stress should be the same as the slope of flow stress, i.e., derivative of flow stress with respect to the true strain, at the necking point. We denoted the true strain and true stress at the necking point as $\varepsilon_{tN}$ and $\sigma_{tN}$, respectively.

The necking in the Swift model occurs at the same engineering strain, regardless of the values of $n_{s}$ and $S_{1}$, when the Considère condition is satisfied for the fixed $\varepsilon_{tN}$. If the flow stress at the selected point before necking must be precisely calculated, $S_{1}$ and $n_{s}$ are uniquely determined. It should be noted that the sensitivity to the strain at the selected point when calculating $S_{1}$ for Swift model is very high. It was previously shown [37] that the flow stress of a typical strain hardening material described by the Hollomon model may be significantly underestimated at high strain after the necking point. In the case that the Considère condition is satisfactory, the strain hardening exponents of Ludwik and Swift models are greater than that of the Hollomon model. In many cases of typical strain hardening materials, the Hollomon model exhibits lower strain hardening behaviors than the actual ones in the case that the exponent was calculated by the Considère condition.

Recently, Jee et al. [123] presented the differences among the fundamental four flow models in terms of the tensile test, described by a black solid line in the lower panel of Figure 14. They fitted the flow stress-strain curves using three conditions of yield strength (305 MPa) at zero strain, tensile strength (engineering stress, 356 MPa), and the Considère condition at the necking point (engineering strain, 0.135) for the Ludwik, Voce, and Swift models and two conditions of tensile strength and the Considère condition at the necking point for the Hollomon model. They thus fitted the flow curves for describing the pre-necking strain hardening behavior of this example. The material constants are listed in Table 2.

The upper four curves in Figure 14 show the four fundamental flow models together with a reference flow stress curve (RFS), which was obtained by Joun et al. [16]. Its corresponding experimental and predicted tensile tests were depicted by solid and dashed lines in the lower panel of Figure 14, respectively. The RFS predicted the tensile test almost exactly because the maximum error was less than 0.4% over the whole range before the fracture starting point denoted by S in Figure 14. Because the flow stresses were calculated for the pre-necking strain hardening behavior, even though the Considère condition was satisfied, the fitted flow stresses and predicted tensile tests are quite different from the reference flow stress (denoted by RFS) and experimental tensile test after the necking point, respectively. Please note that the predicted tensile test by the RFS obtained by Joun et al.’s method [16] is almost exact, as shown in Figure 14. The comparison of the true stress strain curves in the upper panel of Figure 14 reveals that they are quite different from each other. It is also revealed that all curves are not appropriate to meet both the pre-necking and post-necking regions at once for such the material characterized by the employed experimental tensile test even though the Ludwik method came close to the reference flow stress curve.

Figure 15 shows the histories of plastic deformation of the tensile specimens predicted using the four fundamental flow models, revealing that the radii of the necking differ much from each other. This implies the high sensitivity of flow stress obtained depending on the tensile test.

Similarly, with the pre-necking strain hardening emphasized case, we calculated the material constants for the four fundamental material models using two or three conditions of the stress and Considère condition at the necking point and/or the flow stress of RFS at the selected point Q (0.753, 556 MPa). The material constants are listed in Table 3 and their corresponding flow stresses and predicted tensile tests were compared with the RFS and experiments in the upper and lower panels of Figure 16, respectively. The results revealed that all the fitted flow stresses except Hollomon are close to the RFS curve after the true strain of 0.116 at the necking point. Figure 17 shows the predicted tensile tests corresponding to the flow stresses in Figure 16, indicating that they are all acceptable except the Hollomon model, even though the Voce curve is relatively erroneous compared to the other curves.

It is interesting, however, to note that the yield strength $Y_{0}$ fitted by the Ludwik (310 MPa) was acceptable (error 1.6%) while the errors of Swift (338 MPa) and Voce (344 MPa) were 10.8 and 12.8%, respectively. All the fitted yield strengths are greater than the given value of 305 MPa, implying that the accurate yield strength cannot be met even using the scheme of blending two models based on interpolation. This is one of the drawbacks of the four fundamental flow models, because the flow stress of the example cannot be described by them for both the pre- and post-necking strain hardening behaviors at once, even though its pattern is typically monotonic.

For the example employed in this section, which is a typical strain hardening case, it was shown that the Hollomon is not scientific. This may be true of most other cases. Notably, the Ludwik model is superior for this example. In actual engineering activities, however, the Hollomon model is prevalent. Engineers and researchers prefer the Swift model to the Ludwik model even though the latter is easily extendable to thermoviscoplastic materials, as many researchers have already shown. It is presumed that these trends were habituated because the pioneering researchers in this field started to use them in earlier years when their characteristics were not disclosed in terms of tensile test, for example, with its FE predictions and high accuracy [37].

The accuracy of flow stresses of both the pre- and post-necking strain hardening behaviors is important especially in drawing and sheet or plate metal forming processes as well as failure analyses and the statistics on the research activities in this field show that the importance is still growing. It is, however, noted that the traditional fundamental flow models cannot cover the whole strain range of interest and that detailed understanding of their characteristics is important.

### 4.2. Extended Flow Models at Room Temperature

Because the Hollomon and Swift models can be simply derived from the Ludwik model with some assumptions, we called the Ludwik and Voce models the mother models, while the other models were called extended models, even though their mechanical or metallurgical backgrounds might be independent of the mother models.

There are various modifications of the two mother models, Ludwik and Voce, for specific flow descriptions, used by many researchers. In this section, we are going to confine our reviewing range only to the modeling of strain hardening phenomena. Even though the model was targeted to cover strain rate and/or temperature effects as well as strain hardening behavior, we dealt with the model after neglecting the effects of strain rate and temperature. When the equation of the shrunk model was similar to a mother model, we then classified it as a member of the mother model without detailed expression. We summarized the strain hardening models which have the backgrounds of Ludwik and Voce models in Table 4 and Table 5, respectively.

Ludwik family models including Ludwik, Hollomon, and Swift models are well-natured to describe the typical strain hardening behaviors of metallic materials. As a consequence, their modifications from the viewpoint of strain hardening have been rarely made even though they have decisive defects in describing approximately the flow behavior of material to predict the tensile test with an acceptable accuracy, as we stated this fact in the previous section. Table 3 summarizes the variants of the Ludwik family models.

The Ramberg–Osgood model (1943) [124] was presented before the Hollomon (1945) and Swift (1952) models. They stated the total strain as addition of elastic and plastic parts following Hooke’s law and power law, respectively. Therefore, the Hollomon and Swift models can be considered to be approximations of the Ramberg–Osgood model. Ludwigson [82] suggested an empirical model to compensate for the Ludwik model with the drawbacks occurring at low strain in the case of stable austenitic stainless steels and other FCC metals with low stacking fault energies. Ghosh [30] presented the generalized physics-based constitutive model, which is similar to the power law relationship between the stress and plastic strain over wide ranges of strain rates and temperatures. Hartley [125] proposed a generalized Ludwik flow model by expanding the exponential of the flow equation resulting from an incremental constitutive relation [87] to first order.

Joun et al.’s model [16] is defined by the discrete strength coefficient values of Hollomon’s law at the strains of the sample points, which are linearly interpolated to calculate the strength coefficient at any strain. Therefore, is it flexible to describe any kind of flow curve. For example, the flow curve of ESW 105 in Figure 1 was described by this model, which exhibits an almost perfectly plastic or strain softening behaviors. We classified the Kim–Tuan model as one of Ludwik family models because Kim et al. [88] applied it to the cases of very large $V_{2}$ parameters in which Voce’s term is influential in the very small strain. Kim et al. focused on the strain hardening behavior in tensile test of sheet material before necking. The Kim–Tuan model was applied to Al6061-T4, DP980 and CP-Ti and its superiority was shown compared to the Swift and Voce models. However, $V_{2}$’s of their results, i.e., 47.3 for Al6061-T4 and 855 for CP Ti, say that it is not an improved Voce model but an improved Swift model if we focused strictly on their motivation. Recently, Razali et al. [35] proposed a modified Swift model to cope with the limitations of the traditional fundamental flow models.

It is noteworthy that the Ludwik model is the most flexible and thus it has contributed as the basis of the various extended flow rules at the elevated temperatures. Basically, many flow models which cover wide ranges of state variables are thus based on the Ludwik model. These models will be introduced in the next section.

**Table 4 materials-15-02751-t004:** Ludwik family models focused on strain hardening.

No.	Model Name	Model Equation	Reference
1	Ludwik	$\sigma =Y_{0}+L_{1}\varepsilon^{n_{L}}$	[66]
2	Ramberg-Osgood model	$\varepsilon (\sigma )\;\;=\;\;\frac{\sigma }{E}+{\left(\frac{\sigma }{K}\right)}^{1/n}$	[124]
3	Hollomon	$\sigma =L_{1}\varepsilon^{n_{L}}$	[68]
4	Swift	$\sigma =Y_{0}{\left(1+S_{1}\varepsilon \right)}^{n_{S}}$	[69]
5	Ludwigson	$\sigma =K_{1}\varepsilon^{n_{1}}+K_{2}e^{n_{2}\varepsilon }$	[82]
6	Ghosh	$\sigma =A{\left(\varepsilon +B\right)}^{n}-C$	[30]
7	Hartley	$\sigma =Y_{0}+K{\left(\varepsilon +\varepsilon_{0}\right)}^{n}$	[125]
8	Joun et al.	$\sigma =K_{1}(\varepsilon )\;\;\varepsilon^{n_{1}}$ $K_{1}(\varepsilon )=A\;\mathrm{piecewise}\;\mathrm{linear}\;\mathrm{function}$	[16]
9	Kim-Tuan model	$\sigma =Y_{0}+V_{1}(1-e^{-V_{2}\varepsilon }){\left(1+S_{1}\varepsilon \right)}^{n_{S}}$	[88]
10	Razali et al.	$\sigma =Y_{0}{\left(1+\alpha_{1}\varepsilon +\alpha_{2}\varepsilon^{n_{2}}\right)}^{n_{1}\prime }$	[35]

Please note that all the flow constants of the flow models in Table 4 can be referred to in the citation provided.

**Table 5 materials-15-02751-t005:** Voce’s family flow model with an emphasis on strain hardening.

No.	Model Name	Model Equation	Reference
1	Voce	$\sigma =Y_{0}+V_{1}\left(1-e^{-V_{2}\;\varepsilon }\right)$	[67]
2	Bergstrom	$\sigma =Y_{0}+V_{1}{\left(1-V_{3}e^{-V_{2}\varepsilon }\right)}^{0.5}$	[96]
3	Hockett–Sherby	$\sigma =Y_{0}+V_{1}\;(1-e^{-V_{2}\;\varepsilon^{H_{1}}})$	[92]
4	Voyiadjis–Abed	$\sigma =Y_{0}+V_{1}{\left(1-e^{-V_{2}\;\varepsilon }\right)}^{0.5}$	[97]
5	El-Magd	$\sigma =Y_{0}+V_{1}(1-e^{-V_{2}\varepsilon })+E_{1}\varepsilon$	[94]
6	Modified Hockett–Sherby	$\sigma =Y_{0}+V_{1}\varepsilon +V_{2}\left(1-e^{-V_{3}\;\varepsilon^{H_{1}}}\right)$	[18]
7	Double-Voce	$\sigma =Y_{0}+V_{1}\left(1-e^{-V_{2}\;\varepsilon }\right)+V_{3}\left(1-e^{-V_{4}\;\varepsilon }\right)$	[18]
8	Voce generalized	$\sigma =Y_{0}+V_{1}(1-e^{-V_{2}\varepsilon })(1+V_{3}\varepsilon )$	[95]

Please note that all the flow constants of the flow models in Table 5 can be referred to in the citation provided.

Voce’s model was initially developed to describe the material softening phenomenon owing to the dynamic recrystallization. It is thus characterized by the asymptotic stress which constrains the peak stress. When only the strain hardening part is employed to describe the traditional strain hardening behaviors, it has a distinct limitation. To deal with this problem, various modifications were added to its strain hardening part for describing some specific flow behaviors, as shown in Table 5, which summarizes the Voce’s family flow models specialized at room temperature or with strain rate and temperature effects neglected.

Hockett-Sherby [92] targeted the larger strain, for example, of 4.8 for one of their applications, and focused on the materials with large ratio of asymptotic stress to yield strength, for example, ranging from 7.4 to 10.5 for their applications. They did not thus employ the features of Voce’s flow model of the asymptotic stress. In other words, they were interested in the case of extremely small value of $V_{2}$, as can be seen in Figure 18, where the normalized stress is defined as
(5)$$\sigma =\frac{Y_{0}+V_{1}\left(1-e^{-V_{2}\varepsilon }\right)}{Y_{0}+V_{1}}$$

El-Magd et al. [94] used the dislocation evolution model to take into account the strain effect on flow stress of thermoviscoplastic materials. Solving the dislocation evolution equation after some assumption, he formulated a variant of the Voce model, which is the Voce model added to a linear term of strain. El-Magd et al.’s model described accurately the flow stresses at small strain (<0.5) and small strain rate (0.001/s) of C6CrNi18-11 (23 °C), AA7075 T351 (100 °C) and Ck45N (23 °C). It was useful in describing the flow stress of Ck45N at low strain rate, which exhibited the maximum curvature at the strain around 0.05 and linear dependence of flow stress on strain up to the strain of 0.5. It used the features of the Voce model, i.e., the maximum curvature and asymptotic stress at once. However, this model was focused on modeling the thermoviscoplastic materials in which relatively low strains are of great importance. It is known that a linear term cannot be used in modeling the flow stress for covering a wide range of strain. Therefore, this model could not solve the problem of the original Voce model from the standpoint of cold metal forming.

Omer et al. [95] observed the fact that the flow stress of AA7075 varied linearly with strain after a certain degree of plastic deformation at the fixed temperature and thus assumed the $V_{1}$ in Voce model equation as a linear function of strain to describe its flow stress at low strain for its application to hot stamping of AA7075. Therefore, the generalized Voce model has a limitation in describing the flow stresses at small and high strains simultaneously. Voyiadjis and Abed [97] divided the obstacles against movement of dislocations into two parts, i.e., short-term and long-term obstacles. They formulated the long-term obstacle induced flow stress as a modified Voce or Ludwik model. It was revealed that this model was beneficial in the case that strain hardening was concentrated on the small strain. Bergstrom [96] developed a dislocation model covering various dislocation mechanisms of the polycrystalline α-Fe to describe its flow behavior. The dislocation density, formulated by a function of strain, was linked to a part of flow stress. However, they focused on the flow stress before necking point in tensile test. Zhao et al. [18] presented two variants of Voce model, i.e., a modified Hockett–Sherby model and a Double-Voce function model, revealing that the former best fitted HLSA350 and Al6061 while the latter did Q195.

It should be noted that the normalized Voce flow curves in Figure 18 do not represent the flow curves and that all the variables in Equation (5) were found from the related literature. The purpose of these curves is to check the role of the Voce part in the Voce family flow models. It is interesting to summarize that all but the applications of Hockett-Sherby model exploited the Voce model for describing sharply increasing flow patterns at quite small strain from the standpoint of bulk metal forming.

The above literature survey made us conclude that all the Voce model and its variants were developed to meet some specific situations even though they are based on the metallurgical backgrounds. However, the phenomenological observation and its use are still insufficient. In other words, research works for employing the fundamental characteristics of Voce’s strain hardening model can be hardly found.

Each model has its own strong point but all the basic models have some defects in terms of the tensile test which is the simplest plastic deformation problem. Some attempts to employ the strong points of two models at once have been made, based on the blending technique based on the interpolation. However, it is not easy to find a good couple for the specific purpose of satisfying the necking point and pre-necking and post-necking strain hardenings at the same time. For example, it is impossible to find a couple for good blending that solves the matter of describing accurately flow stress of SWCH10A, RFS curve in Figure 14, considering both before and after necking point at once using a blended closed-form function flow model. An extrapolation method may be applied at a little expense of flow behaviors at both sides when the necking condition is satisfied. As mentioned previously, it is because all the initial yield stresses, i.e., yield strengths of the four fundamental flow models, calculated with an emphasis on the necking point and a post-necking strain, are greater than the yield stress of 305 MPa in this case.

Several blended flow models listed in Table 6 were proposed. The combined Hollomon-Voce flow model was proposed by Sung et al. [126] for the more Hollomon-like behavior at the lower strain or vice versa. Banabic and Sester [127] studied the Swift-Hockett/Sherby combined flow model to describe the yield criteria using an additional material parameter of biaxial anisotropy coefficient together with the yield strength, exhibiting an accurate prediction of the sheet metal forming process. Lemoine et al. [128] extrapolated the post-strain hardening behaviors considering the Considère condition. They used the Hollomon, Ludwik, Swift, Voce and Swift–Voce blended flow models and compared the results, showing the superiority of the Swift–Voce blended flow model in terms of the upsetting test and hydraulic bulge test.

The blending technique can be a solution of the problem of the fundamental four flow models stated in the previous section. However, to find a good matching couple is not an easy matter in the case of the example in the previous section, because all the fundamental flow models overestimated the yield strength when the post-necking strain hardening was acceptably described.

## 5. Strain Rate and Temperature Effects on Flow Stress

### 5.1. Limitation of General Closed-Form Function Flow Model

During cold forming, the temperature rise usually softens the flow behaviors while the strain rate usually hardens them. These phenomena affect considerably the flow pattern of the cold metal forming processes, as emphasized in Section 2. However, the dependence of flow behaviors on the strain rate and temperature and its effect have not been much revealed, compared to warm or hot metal forming processes. As a consequence, most research and engineering works are still based on the assumption of negligible effect of these behaviors on the flow stress.

Of course, many general flow models have been presented, which were aimed to cover the wide range of state variables including strain, strain rate, and temperature. A majority of such flow models are based on the Ludwik model, including those by Johnson and Cook [79], Khan et al. [80,81], Samantaray et al. [129,130], Shin and Kim [131], Gao and Zhang [132], Voyiadjis and Almasri [84], Zerilli and Armstrong [83], modified Zerilli-Armstrong [133], etc.

It is noted that it is particularly difficult to find such models that were used to analyze the cold forming processes. This is owing to a distinct limitation that the closed-form function with limited number of material constants is insufficient to describe the flow behaviors introduced in Section 2. It is a fact that the number of material constants can be much increased by formulating the material parameters as functions of state variables. However, this may damage its practicability. It is thus concluded that the general closed-form function flow model is not applicable to cold metal forming simulation.

Table 7 summarizes a few flow models for describing thermoviscoplastically the behavior of metallic materials, which can be applied to thermoviscoplastic FE analyses of cold forming processes after some improvement.

### 5.2. C-m Model

Traditionally, the following *C-m* model has been employed for describing the strain rate effect on flow behaviors at the elevated temperature [139,140]:(6)$$\sigma =C{\dot{\varepsilon }}^{m\;}$$
where *C* is the strength parameter and *m* is the strain rate sensitivity. They can be formulated as functions of strain and temperature as follows:(7)C=C(ε,T) and m=m(ε,T)

It is noteworthy that the model can deal with flow dependence on strain or temperature by formulating *C* and *m* as functions of them. Most flexible description of the *C* and *m* functions is the piecewise bilinear function of the state variables. We can fix m at zero to remove the flow dependence on strain rate.

*C* and *m* are commonly considered to be constants, especially in isothermal analyses [141]. In non-isothermal analyses, *C* and *m* have traditionally been modeled as piecewise bilinear functions of the strain and temperature. When *C* and *m* are described piecewise on a rectangular mesh of strain and temperature, as shown in Figure 19, their values at an arbitrary point A* (ε,*T*) inside a patch, represented by ϕ(ε,T), can be bilinearly interpolated by the following function with the known nodal values of *C* and *m*, denoted by $\phi_{i}(i=1,2,3,4)$ and defined at the grid points:
(8)$$\begin{array}{c} \phi =\frac{\left(T_{j+1}-T\right)\left(\varepsilon_{i+1}-\varepsilon \right)}{\left(T_{j+1}-T_{j}\right)\left(\varepsilon_{i+1}-\varepsilon_{i}\right)}\phi_{1}+\frac{\left(T_{j}-T\right)\left(\varepsilon_{i+1}-\varepsilon \right)}{\left(T_{j}-T_{j+1}\right)\left(\varepsilon_{i+1}-\varepsilon_{i}\right)}\phi_{2}\\ \;\;\;\;+\frac{\left(T_{j+1}-T\right)\left(\varepsilon_{i}-\varepsilon \right)}{\left(T_{j+1}-T_{j}\right)\left(\varepsilon_{i}-\varepsilon_{i+1}\right)}\phi_{3}+\frac{\left(T_{j}-T\right)\left(\varepsilon_{i}-\varepsilon \right)}{\left(T_{j}-T_{j+1}\right)\left(\varepsilon_{i}-\varepsilon_{i+1}\right)}\phi_{4} \end{array}$$

An application example of describing the flow stress of S25C to be cold forged [102], exhibiting a kind of dynamic strain aging phenomena, is introduced. The solid lines in Figure 20 show the given flow stress information. It was described by the *C-m* model and the material constants are listed in Table 8. The fitted flow stresses are denoted by the dot marks in Figure 20. Figure 20 compares the experimental (solid lines) and fitted (dot marks) flow stresses; the results were in good agreement over the entire range of the state variables of interest. The maximum error, of 4.1%, occurred at a strain of 0.4, and at the sample strain rate and temperature of 10/s and 500 °C, respectively.

It is interesting to note that the flow stress curves in Figure 20 are a different description of the flow stress curves in Figure 10. An advanced scheme of *C-m* model [144] aimed at the accurate description of strain rate effect on flow behaviors can also be employed as an improved practical flow model for accurate non-isothermal analyses of cold metal forming processes.

## 6. Conclusions

We investigated the effect of state variables including strain, strain rate, and temperature in the range of cold forming on flow behaviors. In addition, the effects of flow behaviors on the macroscopic phenomena occurring in metal forming were revealed in detail, using typical examples found in the literature. It was emphasized that the tensile test and the Considère condition are particularly important for flow characterization of metallic materials in cold forming.

After reviewing various experimental methods of acquiring flow behaviors, we analyzed the four basic flow models, Ludwik, Hollomon, Voce, and Swift, in terms of a tensile test of a material exhibiting a typical strain hardening behavior. The theoretical flow stress of the tensile test was obtained using Joun et al.’s method [16], which predicted the exact tensile test in the engineering sense. This flow stress was fitted by the four basic flow models with an emphasis on the Considère condition in two ways, i.e., one for the pre-necking region and the other for the post-necking region. The comparison of these predictions revealed that the four basic flow models are not appropriate for describing both the pre- and post-necking strain hardenings with high accuracy simultaneously. Notably, we could not find a matching couple among the four basic flow models which can be simply blended to describe the pre- and post-necking strain hardening behaviors of the material at once using the blending technique based on the interpolation scheme.

We conducted a literature survey on the research works, which aimed at enhancing the problems of the four basic flow models. Their flow models were classified into two categories including Ludwik and Voce flow model families. The backgrounds of the well-known flow models for the materials to be cold formed were explained in detail. The thermoviscoplastic material models were also introduced, which can be employed for non-isothermal analyses of cold forming processes. In particular, we emphasized the importance of traditional piecewise description of flow models; for example, the C-m model used for the CAE applications because it can be flexible, accurate, and practical.

We reviewed and criticized the four fundamental flow models together with their variants from the standpoint of accuracy and application. This criticism became possible because we could identify the post-necking strain hardening behaviors with high accuracy and predict an exact tensile test in the engineering sense, which will improve flow characterization based on full understanding of the macroscopic phenomena of the tensile test.

The problem of flow characterization is extremely complicated because of the strong effect of initial mechanical and metallurgical conditions of test specimens as well as the strong correlation between damage accumulation, microstructural evolution, and tribological effect. A great deal of research works on both flow behavior acquisition and its mathematical modeling should be thus made for the enhancement of accuracy and practical application to precision FE analyses of metal forming processes.

## Figures and Tables

**Figure 1 materials-15-02751-f001:**
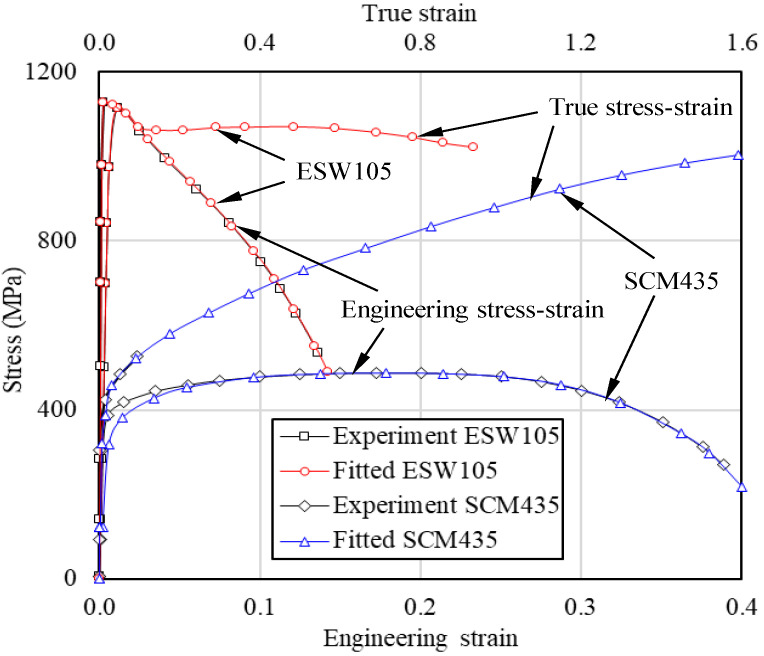
Engineering and true stress-strain curves of ESW105 and SCM435 [4].

**Figure 2 materials-15-02751-f002:**
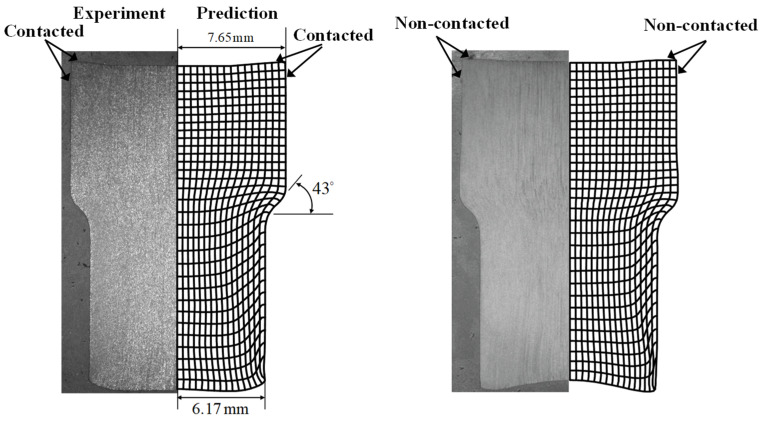
Comparison of the deformed shapes between the experiments and predictions in axisymmetric forward extrusion [4].

**Figure 3 materials-15-02751-f003:**
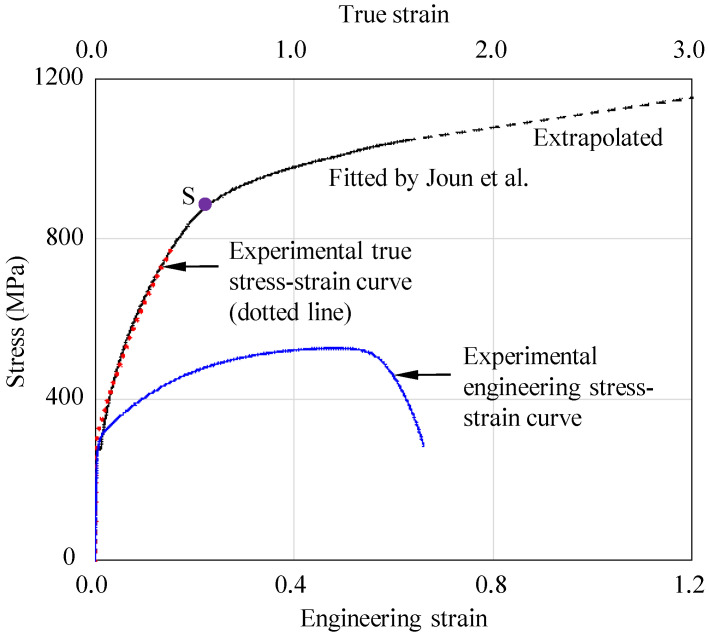
Tensile test curves and the flow stress curve of SUS304 at room temperature [8].

**Figure 4 materials-15-02751-f004:**
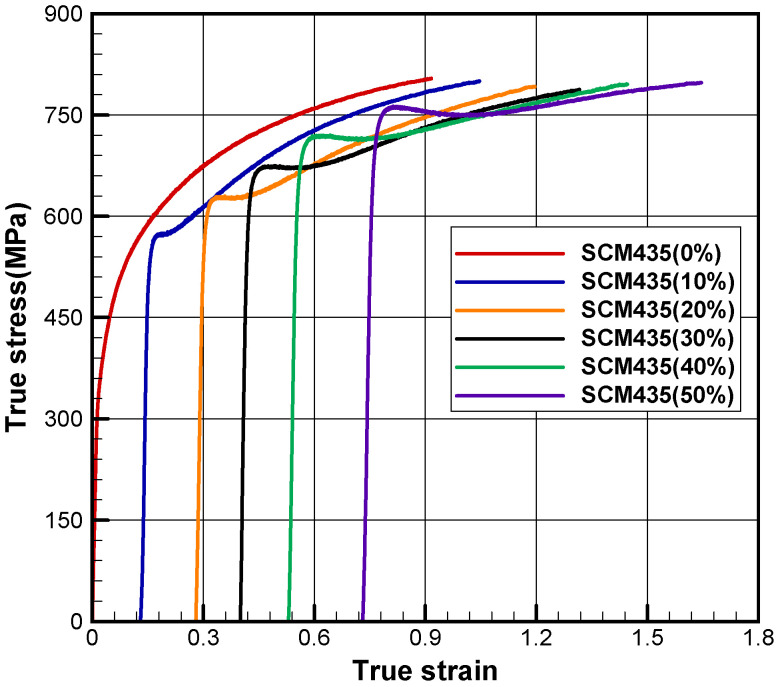
Bauschinger effect occurring in cylinder compression test of the drawn specimens [65].

**Figure 5 materials-15-02751-f005:**
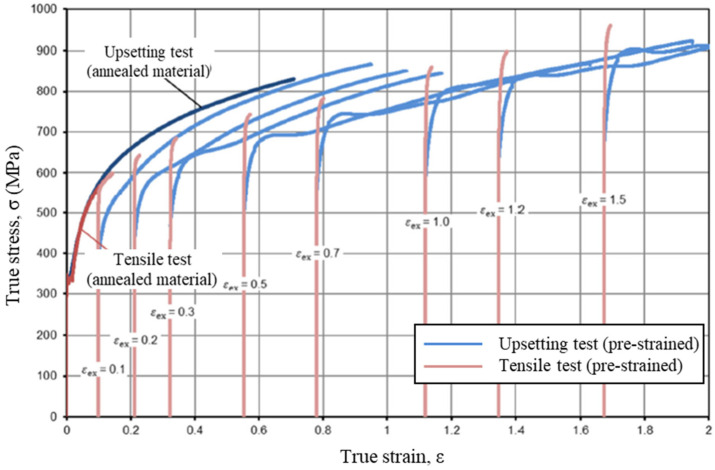
Bauschinger effect occurring in tensile test of the upset specimens [28].

**Figure 6 materials-15-02751-f006:**
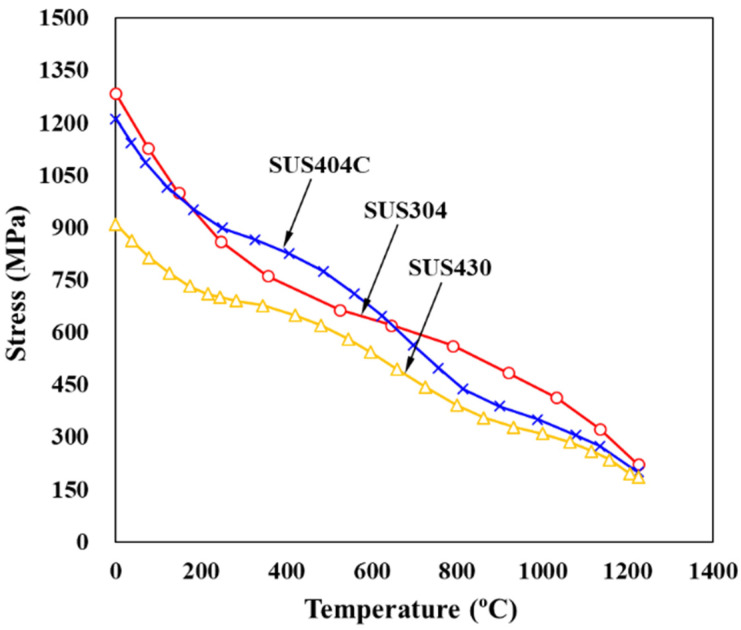
Dynamic strain aging of various stainless steels [105].

**Figure 7 materials-15-02751-f007:**
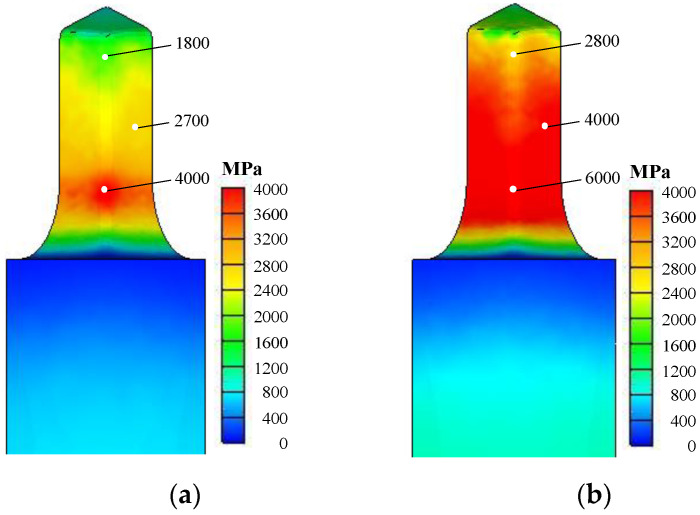
Dies stresses obtained by isothermal and non-isothermal FE analyses [8]: (**a**) Non-isothermal; (**b**) Isothermal.

**Figure 8 materials-15-02751-f008:**
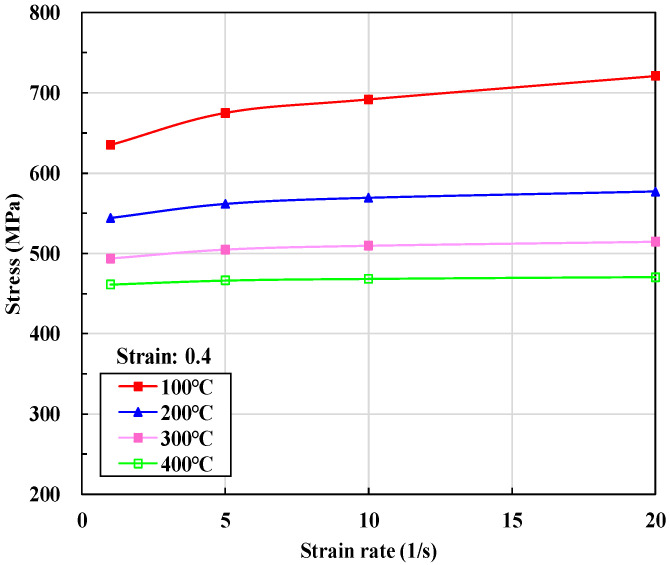
Changes in flow stress by strain rate at various sample strains and temperatures [102].

**Figure 9 materials-15-02751-f009:**
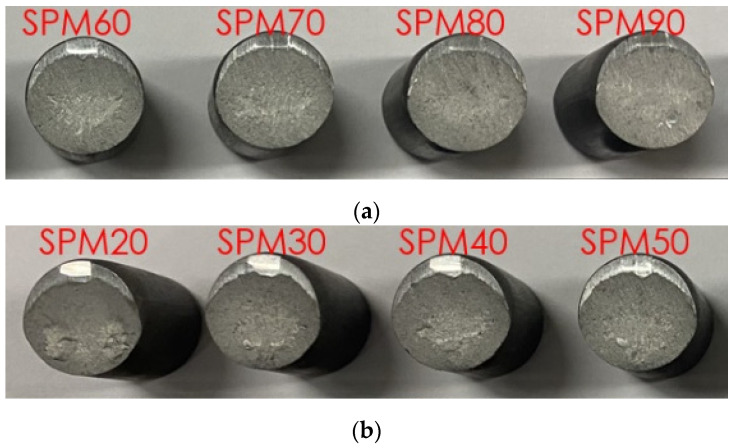
Effect of speed of rod shearing in automatic multi-stage cold forging on the quality of the sheared surface: (**a**) Good sheared surface; (**b**) Bad sheared surface. SPM stands for “strokes per minute”.

**Figure 10 materials-15-02751-f010:**
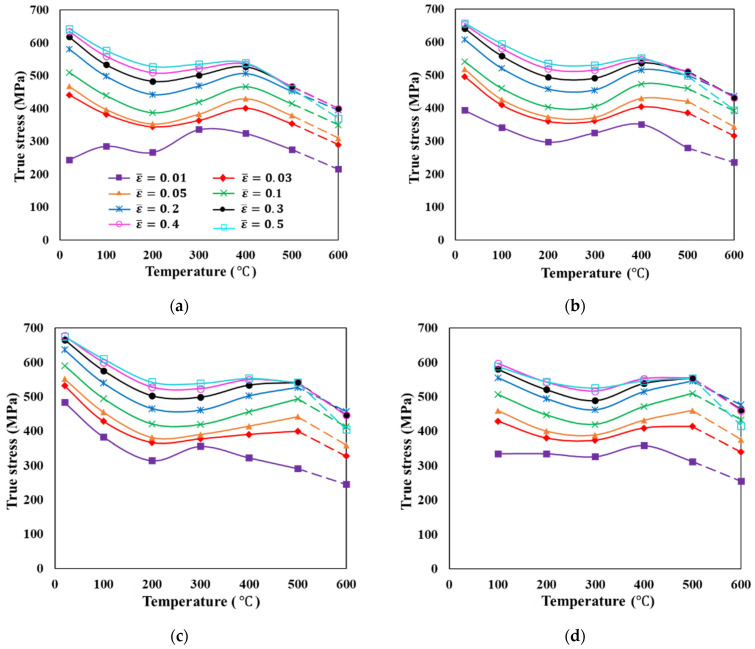
Variation in flow stress with temperature at strain rates of: (**a**) Strain rate: 1/s; (**b**) Strain rate: 5/s; (**c**) Strain rate: 10/s; (**d**) Strain rate: 20/s.

**Figure 11 materials-15-02751-f011:**
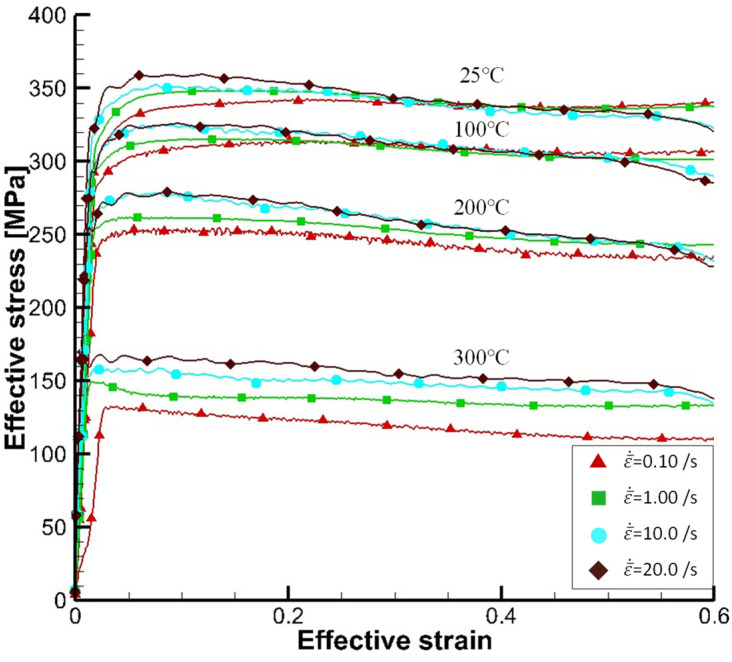
Effective strain-flow stress curves [106].

**Figure 12 materials-15-02751-f012:**
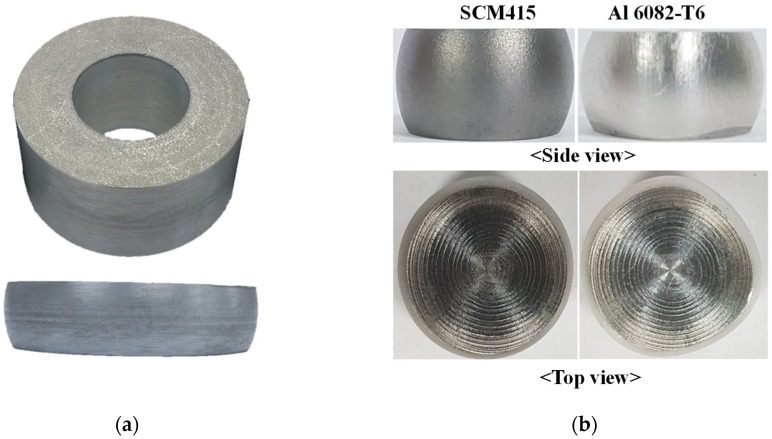
Instability of aluminum alloys during upsetting: (**a**) Ring compression; (**b**) Solid cylinder compression.

**Figure 13 materials-15-02751-f013:**
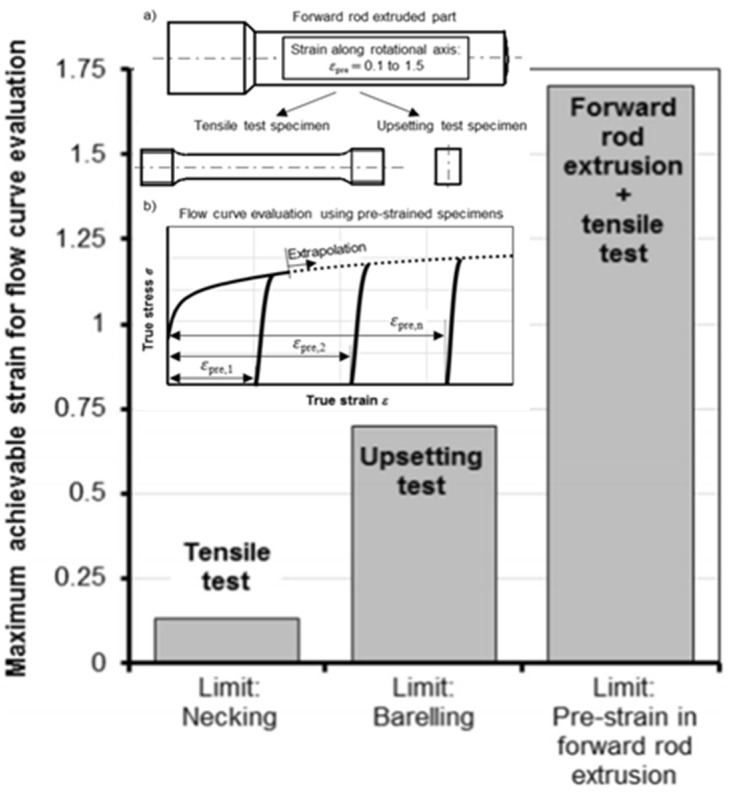
Strain limit of testing methods in calculating the flow stress [28]: (**a**) Test specimen; (**b**) Flow curve evaluation using pre-strained specimens.

**Figure 14 materials-15-02751-f014:**
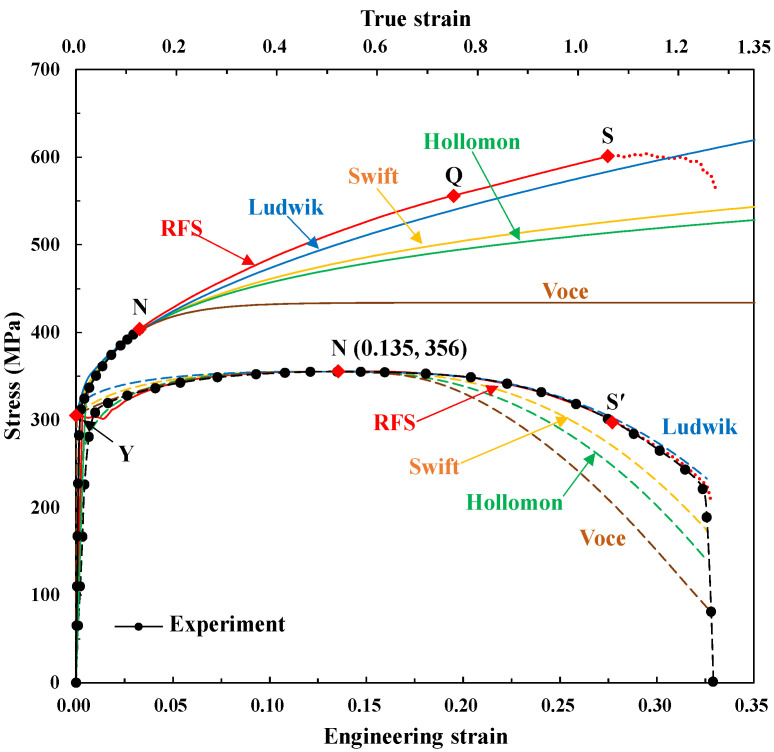
Comparison of flow stresses and their corresponding tensile test predictions of the four fundamental flow models with an emphasis on necking point and yield strength at zero strain.

**Figure 15 materials-15-02751-f015:**
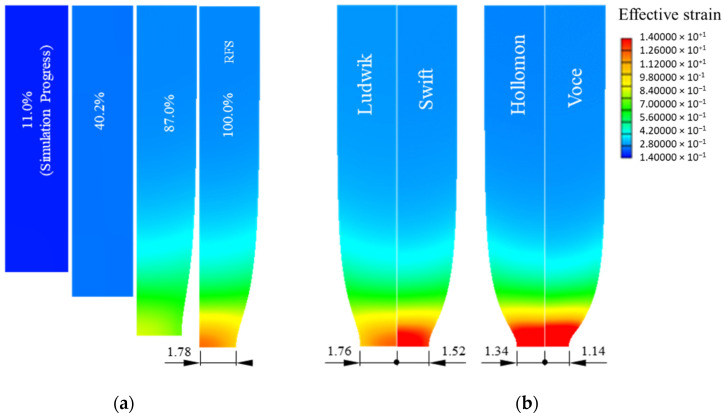
Comparison of deformed shapes of the tensile test, predicted using the RFS and fundamental flow models: (**a**) RFS; (**b**) Fundamental flow models.

**Figure 16 materials-15-02751-f016:**
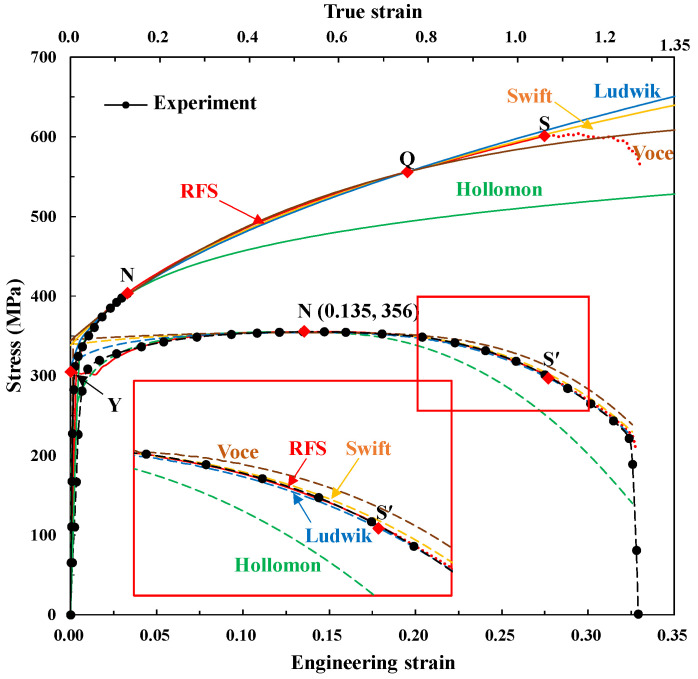
Comparison of flow stresses and their corresponding tensile test predictions of the four fundamental flow models with an emphasis on necking point and a selected point on the RFS curve.

**Figure 17 materials-15-02751-f017:**
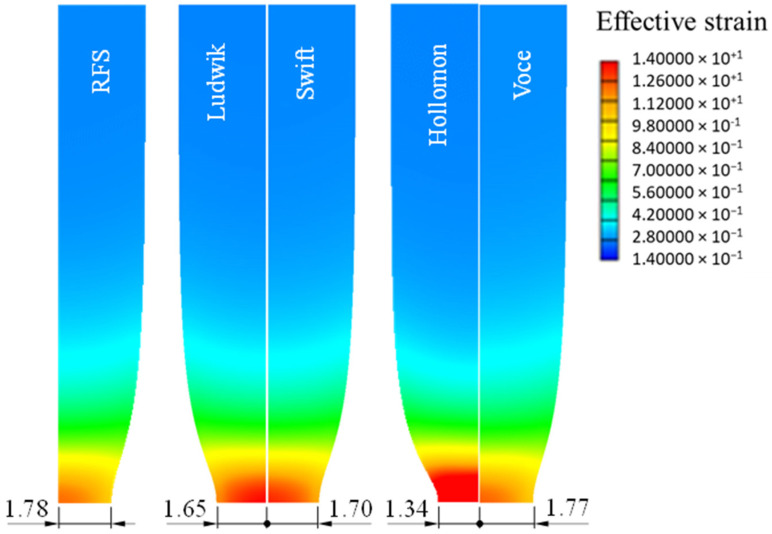
Comparison of deformed shapes of the tensile test, predicted using the RFS and fundamental flow models.

**Figure 18 materials-15-02751-f018:**
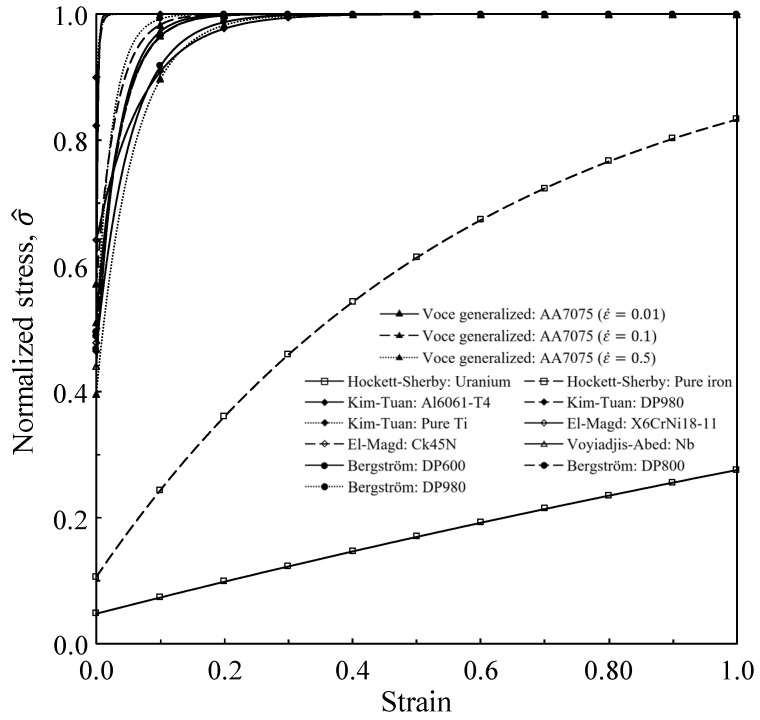
Normalized Voce curves.

**Figure 19 materials-15-02751-f019:**
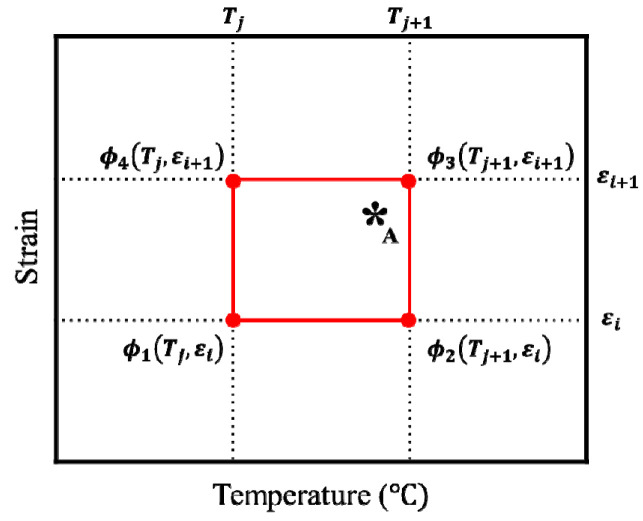
Piecewise bilinear expression of material constants using the sample grid points in the form of *ϕ* (strain, temperature) [142,143].

**Figure 20 materials-15-02751-f020:**
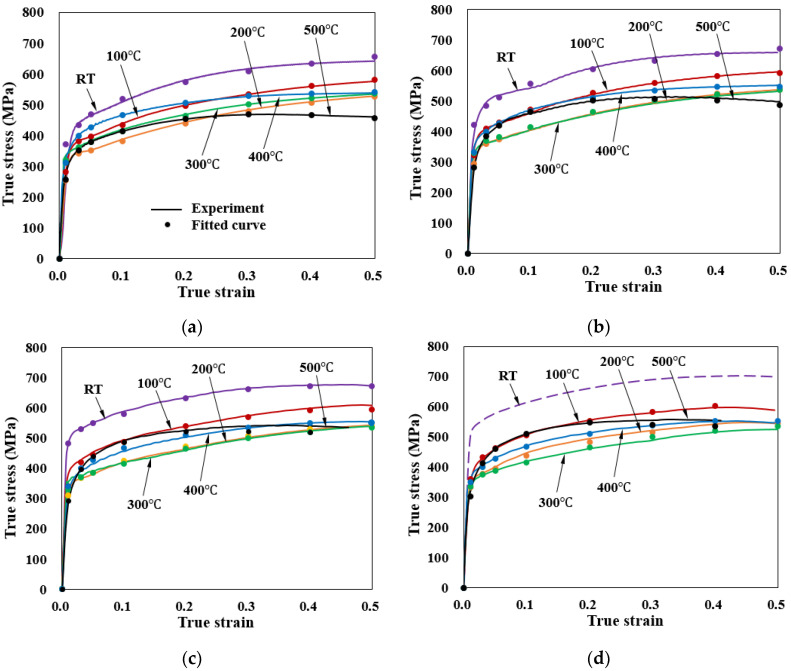
Experimental and fitted flow stresses with respect to strain and temperature, for various strain rates [102]: (**a**) 1.0, (**b**) 5.0, (**c**) 10.0, and (**d**) 20/s.

**Table 1 materials-15-02751-t001:** Methods of obtaining flow behaviors at room temperature and its strain limits.

Researchers	Method	Specimen Type	Strain Limit
Bridgman et al. [12]	Tensile test	Solid cylinder	-
Mirone [13]	Tensile test	Solid cylinder	1.1
Martin et al. [14]	Tensile test. Force-area measurement	Sheet	1.5
Boger et al. [20]	Tension/compression	Sheet	0.25
Hwang et al. [31]	Hydraulic bulge	Tube	0.22
Lach, Pöhlandt [26]	Torsion	Solid cylinder, tube	-
Bouvier et al. [27]	Simple shear	Rolled sheet	0.5
Kuwabara [29]	Cruciform–biaxial	Sheet	1.1
Smith et al. [21]	Sheet bulging	Sheet	0.55
Merklein, Kuppert [22]	Layer compression	Sheet	0.45
Marciniak, Kolodziejski [24]	In-plane torsion	Ring	-
Tekkaya et al. [25]	In-plane torsion	Sheet	0.8
Hering et al. [28]	Forward extrusion and tensile test	Solid cylinder	1.7
Dziallach et al. [33]	Tensile test, hydraulic bulge test	Sheet	1.0
Kopp, Putten [23]	Multi-layer plane strain compression	Very thin strip	0.8
Kajberg, Lindkvist [15]	Tensile test, FEA	Notched thin sheet	0.8
Joun et al. [16]	Tensile test, FEA	Solid cylinder	1.6
Kamaya, Kawakubo [19]	Tensile test, FEA	Notched solid cylinder	0.85
Zhao et al. [18]	Tensile test, FEA	Notched sheet	0.9

**Table 2 materials-15-02751-t002:** Material constants of the four fundamental flow models focused on pre-necking.

Model Name	Material Constants
Ludwik	$Y_{0}=305\;\;\mathrm{MPa},\;\;\;L_{0}=271\;\;\mathrm{MPa},\;\;\;n_{L}=0.499$
Voce	$Y_{0}=305\;\mathrm{MPa},\;V_{1}=129\;\mathrm{MPa},\;V_{2}=10.9$
Hollomon	K=510 MPa, n=0.116
Swift	$Y_{0}=305\;\;\mathrm{MPa},\;S_{1}=52.4,n_{S}=0.135$

**Table 3 materials-15-02751-t003:** Material constants of the four fundamental flow models focused on post-necking.

Model Name	Material Constants
Ludwik	$Y_{0}=310\;\;\mathrm{MPa},\;\;\;L_{0}=289\;\;\mathrm{MPa},\;\;\;n_{L}=0.557$
Voce	$Y_{0}=344\;\;\mathrm{MPa},\;V_{1}=294\;\;\mathrm{MPa},\;V_{2}=1.69$
Hollomon	K=510 MPa, n=0.116
Swift	$Y_{0}=338\;\;\mathrm{MPa},\;S_{1}=6.72,n_{S}=0.276$

**Table 6 materials-15-02751-t006:** Combined models of flow stress.

No.	Model Name	Model Equation with Description	Reference
1	Sung et al.	$\sigma =\alpha \;K\varepsilon^{n}+(1-\alpha )\;\;\left\{Y_{0}+V_{1}(1-e^{-V_{2}\;\varepsilon })\right\}$ Hollomon and Voce	[126]
2	Banabic and Sester	σ=α{Y01(1+Sε1)n1}+(1−α) {Y02+H1 (1−e−H2 εH3)} Swift and Hockett/Sherby	[127]
3	Lemoine, Sriram, Kergen	σ=αY01(1+Sε1)n+(1−α) {Y02+H1 (1−e−H2 ε)} Swift and Voce	[128]

Please note that all the flow constants of the flow models in Table 6 can be referred to the in citation provided.

**Table 7 materials-15-02751-t007:** Thermoviscoplastic models that can be potentially applied to cold forming.

No.	Model Name	Model Equation	Reference
1	Johnson-Cook	$\begin{array}{l} \sigma =(A+B\varepsilon^{n})\left[1+C\ln\left(\frac{\dot{\varepsilon }}{{\dot{\varepsilon }}_{0}}\right)\right]\left[1-{\left(\frac{T-T_{r}}{T_{m}-T}\right)}^{m}\right]\\ T_{r}=\mathrm{reference}\;\mathrm{temperature}\\ T_{m}=\mathrm{melting}\;\mathrm{temperature} \end{array}$	[79]
2	Khan et al.	$\sigma =\left[A+B\varepsilon^{n_{0}}{\left(1-\frac{\ln\dot{\varepsilon }}{\ln{\dot{\varepsilon }}_{\max}}\right)}^{n_{1}}\right]{\left(\frac{\dot{\varepsilon }}{{\dot{\varepsilon }}_{0}}\right)}^{m}{\left(\frac{T_{m}-T}{T_{m}-T_{0}}\right)}^{\beta }$	[80,81]
3	Ludwigson	$\sigma =K_{1}\varepsilon^{n_{1}}+e^{\left(K_{2}+n_{2}\varepsilon \right)}$	[82]
4	Zerilli and Armstrong	$\sigma =K_{1}\varepsilon^{n_{1}}\left(n_{2}T+n_{4}T\ln\dot{\varepsilon }\right)$	[83]
5	Modified Zerilli and Armstrong	$\begin{array}{l} \sigma =c_{0}+B_{0}e^{-T\left(\beta_{0}-\beta_{1}\ln\dot{\varepsilon }\right)}+K\varepsilon^{n}\;\mathrm{for}\;\mathrm{BCC}\\ \sigma =c_{0}+B_{0}e^{-T\left(\beta_{0}+\beta_{1}\ln\dot{\varepsilon }\right)}\;\mathrm{for}\;\mathrm{FCC} \end{array}$	[133]
6	Fields–Backofen	$\sigma =K_{1}\varepsilon^{n_{1}}{\dot{\varepsilon }}^{n_{2}}$	[70]
7	Modified Fields-Backofen	$\sigma =K_{1}\varepsilon^{n_{1}}{\dot{\varepsilon }}^{n_{2}}e^{\frac{K_{2}}{T}}$	[134]
8	Samantaray et al.	$\sigma =\left(A+B\varepsilon^{n}\right)\left[1-{\left(T-T_{r}\right)}^{m}\right]$	[129,130]
9	Gao and Zhang	$\sigma =\sigma_{a}+Y\varepsilon^{n}e^{\left[c_{3}T\ln\left(\frac{\dot{\varepsilon }}{{\dot{\varepsilon }}_{s0}}\right)\right]\left[1+-c_{4}T\ln\left(\frac{\dot{\varepsilon }}{{\dot{\varepsilon }}_{s0}}\right)\right]}$	[132]
10	Voyiadjis-Almasri	$\sigma =Y_{0}+K_{1}\varepsilon_{p}^{n_{1}}\left[B_{1}T{\left(\varepsilon_{p}\right)}^{\frac{1}{n_{2}}}+B_{2}Te^{n_{3}\left(1-\frac{T}{T_{m}}\right)}+1\right]$	[84]
11	Rusinek-Klepaczko	$\sigma =Y_{0}\left[V_{1}{\left\{\varepsilon_{0}+\varepsilon_{p}\right\}}^{n_{1}}+V_{2}\left\{1-D_{1}\left(T/T_{m}\right)\log\left({\dot{\varepsilon }}^{p}/\varepsilon_{\min}\right)\right\}\right]$	[89,135]
12	Hensel–Spittel	$\sigma =A\varepsilon^{m_{1}}e^{\frac{m_{2}}{\varepsilon }+m_{3}\varepsilon }{\dot{\varepsilon }}^{m_{4}+m_{5}T}T^{m_{6}}e^{\frac{m_{7}}{T}+m_{8}T}{\left(1+\varepsilon \right)}^{m_{9}T}$	[136]
13	Modified Voce	$\sigma =Y_{0}+V_{1}\;{\left[1-V_{2}{\left(\ln\frac{{\dot{\varepsilon }}_{0}}{\dot{\varepsilon }}\right)}^{\frac{1}{A}}\right]}^{\frac{1}{B}}$	[60]
14	Voyiadjis–Abed	$\sigma =c_{1}+c_{2}\sqrt{1-e^{-c_{3}\;\varepsilon }}+c_{4}{\left[1-{\left(c_{5}T-c_{6}T\ln\dot{\varepsilon }\right)}^{\frac{1}{q_{1}}}\right]}^{\frac{1}{q_{2}}}$	[91]
15	Shin–Kim	$\sigma =\left[A+B\left(1-e^{-C\varepsilon }\right)\right]\left[D\ln(\dot{\varepsilon }/{\dot{\varepsilon }}_{0})+e^{E\dot{\varepsilon }/{\dot{\varepsilon }}_{0}}\right]\left[1-{\left(\frac{T-T_{r}}{T_{m}-T_{r}}\right)}^{m}\right]$	[131]
16	Hockett-Sherby	$\sigma =\sigma_{0}+Q\left(1-e^{-b\;\varepsilon^{n}}\right)$	[92]
17	Modified Hockett–Sherby	$\sigma =\left[\sigma_{0}+Q\left(1-e^{-b\;\varepsilon^{n}}\right)\right]{\left(\frac{\dot{\varepsilon }}{{\dot{\varepsilon }}_{0}}\right)}^{m}$	[18,93]
18	Modified Hockett–Sherby	$\sigma =\sigma_{0}+Q{\left[1-{\left(bT\ln\dot{\varepsilon }\right)}^{\frac{1}{q_{1}}}\right]}^{\frac{1}{q_{2}}}$	[137]
19	Cai et al.	$\begin{array}{l} \sigma =c_{1}+c_{3}\mu \sqrt{1-e^{-c_{2}\varepsilon }}+c_{4}\mu {\left[\left(1+q\right){\left(\dot{\varepsilon }/{\dot{\varepsilon }}_{0}\right)}^{c_{5}T}-1\right]}^{1/p}\;\mathrm{for}\;\mathrm{BCC}\\ \sigma =c_{1}+\mu \sqrt{1-e^{-c_{2}\varepsilon }}\langle c_{3}+c_{4}{\left[\left(1+q\right){\left(\dot{\varepsilon }/{\dot{\varepsilon }}_{0}\right)}^{c_{5}T}-1\right]}^{1/p}\rangle \;\mathrm{for}\;\mathrm{FCC} \end{array}$	[138]
20	Hartley et al.	$\sigma =Y_{0}+K{\left(\varepsilon -\varepsilon_{e}\right)}^{n}{\left(\dot{\varepsilon }/{\dot{\varepsilon }}_{0}\right)}^{m}$	[125]

Please note that all the flow constants of the flow models in Table 7 can be referred to the citation provided.

**Table 8 materials-15-02751-t008:** Material constants of *C* and *m* values at the sample points for $\sigma =C{\dot{\varepsilon }}^{m}$ [102].

*ε*	*T* (°C)											
	20		100		200		300		400		500	
	*C*	*m*	*C*	*m*	*C*	*m*	*C*	*m*	*C*	*m*	*C*	*m*
0.01	237.55	0.3063	287.20	0.0780	254.44	0.0882	320.27	0.0165	312.19	0.0387	255.42	0.0586
0.03	436.16	0.0830	380.92	0.0432	341.52	0.0332	360.96	0.0119	397.78	0.0025	351.82	0.0526
0.05	464.93	0.0717	396.05	0.0514	351.29	0.0396	378.64	0.0071	427.68	−0.0031	379.12	0.0638
0.1	504.93	0.0598	434.71	0.0501	381.81	0.0461	415.22	0.0002	466.51	−0.0004	413.53	0.0709
0.2	577.82	0.0385	495.77	0.0365	437.96	0.0337	465.21	−0.0053	507.8	0.0026	453.06	0.0628
0.3	615.25	0.0314	533.70	0.0294	479.74	0.0239	500.56	−0.0068	528.50	0.0062	468.00	0.0585
0.4	632.62	0.0265	560.37	0.0240	506.69	0.0200	520.65	−0.0019	536.40	0.0103	466.22	0.0596
0.5	640.86	0.0206	580.71	0.0115	527.37	0.0109	535.50	−0.0034	542.05	0.0061	457.35	0.0641

## Data Availability

Not applicable.

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
