# Peer review of "A Review of Flow Characterization of Metallic Materials in the Cold Forming Temperature Range and Its Major Issues"

_materials, 2022, doi:10.3390/ma15082751_

Round 1
Reviewer 1 Report
Dear Authors
The article is interesting and provides an overview of many methods and publications. However, it requires a few corrections:
1. Figures 6, 10c and 10d need to be improved.
2. I propose to change the title of the chapter “2. Materials and Methods ”. The article is not based on a research – it is a review. This chapter title does not fit to this type of an article. Typically, such titles are used in a research-based work.
3.Regard Figure 7, what is the reason for such high stresses of 6000 MPa?
Correct the scale results outside the scale range.
Best Regards
reviewer
Author Response
We would like to thank the reviewer for the valuable comments and very helpful suggestions. We carefully revised the manuscript according to the reviewer's advice. Also, we addressed the Reviewer's comments point by point. The revision made in the manuscript is highlighted in yellow and marked up using the “Track Changes” function in word. Please kindly refer to the attached file for the response.

Reviewer 2 Report
Review
The article reviews the literature on the models used to describe cold-deformation material. The study verified the results of the models used against the results of experimental research. Strengths and weaknesses of the material flow models used in, for example, numerical modelling are indicated. The models have been analysed in terms of various strength tests, mainly stretching. The author paid great attention to the narrowing of the samples during stretching. The influence of temperature and strain rate was also analysed.
The subject is interesting and the problem is difficult to generalize. The authors base their knowledge on the literature. A comprehensive list of references can be an interesting source of knowledge for readers.
There are some shortcomings in the work that should be corrected. Detailed comments have been included in the file.
Yours sincerely
Reviewer

Author Response

(The authors gave the same response as above.)

Reviewer 3 Report
The article entitled "A review on flow characterization of metallic materials in the cold forming temperature range with the major issues" presents a comprehensive review about behavior of materials under cold deformation. In my opinion the paper is worth to be published. They provide a large list of reference and discussed research works in a well-structured document. Only minor suggestions are proposed for clarification for readers.
Figure 6: Is this figure correct or something is missed on it?
Figure 10: I think that there is something missed on these figures c and d.
Equations in tables (and text): You should include a definition of each variable of the equations below the equation. Otherwise, it is not clear the meaning of each term.
Author Response

(The authors gave the same response as above.)
